# Effect of Maternal Catalase Supplementation on Reproductive Performance, Antioxidant Activity and Mineral Transport in Sows and Piglets

**DOI:** 10.3390/ani12070828

**Published:** 2022-03-24

**Authors:** Guanglun Guo, Tiantian Zhou, Fengyun Ren, Jingzhan Sun, Dun Deng, Xingguo Huang, Teketay Wassie, Izhar Hyder Qazi, Xin Wu

**Affiliations:** 1College of Animal Science and Technology, Hunan Agricultural University, Changsha 410128, China; ggl01041318@163.com (G.G.); renfengyun@foxmail.com (F.R.); sun-jingzhan@foxmail.com (J.S.); teketay@isa.ac.cn (T.W.); 2Key Laboratory of Agro-Ecological Processes in Subtropical Region, Institute of Subtropical Agriculture, The Chinese Academy of Sciences, Changsha 410125, China; zhoutiantian19@mails.ucas.ac.cn; 3National Engineering Laboratory for Pollution Control and Waste Utilization in Livestock and Poultry Production, Changsha 410125, China; 4Tangrenshen Group Co., Ltd., Zhuzhou 330500, China; dd-yf@trsgroup.cn; 5Engineering Research Center for Feed Safety and Efficient Utilization of Education, Hunan Agricultural University, Changsha 410128, China; 6Department of Veterinary Anatomy and Histology, Shaheed Benazir Bhutto University of Veterinary and Animal Sciences, Sakrand 67210, Pakistan; vetdr_izhar@yahoo.com

**Keywords:** catalase, reproductive performance, antioxidant enzyme, mineral element, placenta, pregnant sow, neonatal piglet

## Abstract

**Simple Summary:**

Oxidative stress negatively affects maternal health and fetal development. Catalase (CAT) is an oxidoreductase enzyme that catalyzes the decomposition of hydrogen peroxide into molecular oxygen and water, thereby protecting the cell from oxidative damage. This experiment was conducted to investigate the effects of maternal CAT supplementation on reproductive performance, antioxidant enzyme activities, mineral transport, and the mRNA expression of related genes in sows and offspring. It was observed that giving dietary CAT supplementation to pregnant sows may decrease the intrauterine growth restriction (IUGR) rate and contribute to improving the maternal and fetal antioxidant status, potentially by modulating the activities of selected antioxidant enzymes and mRNA expression of related genes, as well as the transport of some mineral elements in the pregnant sows and their piglets. This finding provides a reasonable foundation for focused intervention studies in the future and may also provide insight into important aspects of mother-infant nutrition in both animals and humans.

**Abstract:**

This experiment was conducted to investigate the effects of maternal catalase (CAT) supplementation on reproductive performance, antioxidant enzyme activities, mineral transport, and mRNA expression of related genes in sows and offspring. A total of 40 pregnant sows at 95 days of gestation with similar parity (3–5 parities) and back-fat thickness were assigned randomly and equally into the control (CON) group (fed a basal diet) and CAT group (fed a basal diet supplemented with 660 mg/kg CAT; CAT activity, 280 U/g). The reproductive performance was recorded, and the placenta and blood samples of sows and neonatal piglets, as well as the jejunum and ileum samples from neonatal boars (eight replicates per group), were collected. Results showed that dietary supplementation with CAT significantly decreased the intrauterine growth restriction (IUGR) rate and increased the activity of serum CAT in neonatal piglets and umbilical cords (*p* < 0.05). In addition, CAT supplementation tended to improve total antioxidant capacity (T-AOC) levels in the maternal serum (*p* = 0.089) and umbilical cords of piglets (*p* = 0.051). The serum calcium (Ca), manganese (Mn), and zinc (Zn) of farrowing sows and Mn concentration in the umbilical cord, and serum Ca, magnesium (Mg), copper (Cu), and Mn of neonatal piglets were significantly increased (*p* < 0.05) in the CAT group. CAT supplementation downregulated mRNA expression of *TRPV6* and *CTR1* (*p* < 0.05), *Cu/Zn SOD* (*p* = 0.086) in the placenta and tended to increase the mRNA expression of the glutathione peroxidase 1 (*GPX1*) (*p* = 0.084), glutathione peroxidase 4 (*GPX4*) (*p* = 0.063), and *CAT* (*p* = 0.052) genes in the ileum of piglets. These results showed that the maternal CAT supplementation improved fetal growth by decreasing the IUGR rate, and modulated antioxidant activity, as well as mineral elements in the pregnant sows and their piglets.

## 1. Introduction

During late gestation, a significant increase in the rate of fetal and mammary gland growth is observed [1,2], which leads to progressive oxidative stress in sows [3,4]. This is accompanied by a decrease in antioxidant enzymes in sow’s blood during late gestation [5]. Apart from its direct impact on the sows, oxidative stress can also lead to pregnancy complications such as idiopathic recurrent pregnancy loss and embryogenesis defect [6]. Oxidative stress in sows, especially during late pregnancy, can affect the transport of nutrients and oxygen from mother to offspring and can impair fetal development [7]. It follows that oxidative stress reduces the reproductive performance of sows, such as the litter sizes (total born), born alive, and litter weights [8]. Newborn piglets are vulnerable to the stress ascribed to the sudden change revulsion from the endouterine to extrauterine environment with relatively high oxygen. The piglets may be exposed to an overproduction of reactive oxygen species (ROS) and lipid peroxidation, leading to damage to cell membranes and structures [9,10]. It has been reported that lipid peroxidation and antioxidant status are altered during parturition, resulting in oxidative stress in the fetus [11]. Maternal oxidative status has a strong correlation with neonatal stress, high maternal oxidative stress corresponds to impaired placental structure and function, resulting in adverse effects of nutrients and oxygen transported from the placenta to the fetus, which raises neonatal oxidative stress [12].

Living organisms have sophisticated and efficient antioxidant defense systems against ROS, especially antioxidant enzymes, including catalase (CAT), superoxide dismutase (SOD), and glutathione peroxidase (GPX). CAT is a widespread enzyme found in nearly all living organisms exposed to oxygen, which catalyzes the decomposition of hydrogen peroxide (H_2_O_2_) into molecular oxygen and water to protect the cell from oxidative damage. Several studies have shown that CAT plays an important role in regulating growth performance [13,14]. In this regard, dietary supplementation with 2000 mg/kg exogenetic CAT (dietary CAT activity, 120 U/kg) has been reported to improve growth performance and enhance antioxidant enzyme activities in weaned piglets, alleviate oxidative stress, and reduce hepatic damage by suppressing hepatic apoptosis [14]. Mitigation of oxidative stress and associated damage is not only beneficial to animal health and welfare but can also improve their production performance. This notion is supported by the fact that when the energy utilized by the immune system is reduced, more protein and energy are available to promote growth [15]. In addition, mineral elements including copper, magnesium, zinc, and manganese possess antioxidant properties and exert protective effects against oxidative stress [16,17]. It has been reported that the administration of antioxidant enzymes along with essential trace elements and minerals can reduce the extent of oxidative damage [18]. Specific transporters are required for the movement of mineral elements across the phospholipid bilayer into the cell [19]. Therefore, it is reasonable to speculate that dietary CAT supplementation may affect serum mineral elements and transporters.

There is a lack of information regarding the implication of maternal CAT supplementation on reproductive performance and oxidative status in the later phase of pregnancy of sows and their neonatal piglets. Beyond that, the effects of CAT on serum mineral elements and transporters are also less studied. Based on evidence from previous studies [20,21], we hypothesized that maternal CAT supplementation may have other potential health benefits beyond boosting antioxidase activities. Therefore, the present research was designed to study the potential implication of maternal dietary CAT supplementation during late pregnancy on reproductive performance, antioxidant enzyme activities, mineral (Ca, phosphorus (P), Mg, Cu, Mn, and Zn) transport, and mRNA expression of related genes in sows and their neonatal piglets.

## 2. Materials and Methods

### 2.1. Experimental Design, Animals, and Management

Forty pregnant sows (Large White× Landrace) at day 95 of gestation with similar parity (3–5 parities) and back-fat thickness were selected. All sows were randomly assigned to one of the two dietary treatments (20 sows per treatment). The treatments consisted of a basal diet (CON group), or a basal diet supplemented with exogenous CAT (CAT group). The dietary experiment was carried out from the late gestation (day 95) to the end of farrowing. The experiment was carried out in Hunan Tangrenshen Group Co., Ltd. (Yueyang, China). The housing and reproductive management of all experimental sows were carried out as described previously [22].

### 2.2. Diets

Sows in the CON group were fed a basal diet and those in the CAT group were fed the basal diet supplemented with 660 mg/kg CAT (CAT activity, 280 U/g); the CAT was in the form of embedding, which mixed evenly with the basal diet. The corn-soybean-based diet was used in the present experiment (Table 1), and all nutrients of the basal diet were calculated to meet the nutrient recommendation of the NRC 2012 for sows [23].

All sows were fed twice daily at 07:00 and 16:00 h and had free access to water throughout the experiment. The dietary intake of each sow was artificially adjusted to 3.0 kg/d from day 95 to 110 of gestation and 1.8 kg/d for the rest days before farrowing (792 mg/head/day CAT was artificially supplemented to maintain constant intake) [24,25]. This was performed to adjust for the loss of appetite before farrowing and to improve the feed intake of sows after parturition.

### 2.3. Sample Collection

In the course of the experiment, 16 sows (8 per treatment) with similar body conditions farrowed almost simultaneously at day 114 of gestation. These sows had a spontaneous delivery without assistance and exogenous hormone induction. To avoid the influence of delivery time, we selected these 16 sows and their piglets as research objects. One male piglet with a birth weight of 1.30–1.50 kg (mean ± SEM, 1.38 ± 0.213 kg) was selected from each litter as an experimental animal. A total of 8 boars (only) were selected from each group based on previous research [25], to control variables and eliminate the influence of gender. The 16 piglets were divided into two groups: (1) CON group (*n* = 8), (2) CAT group (*n* = 8).

Reproductive performances of sows, such as the litter size (total born), the number of piglets born alive, and intrauterine growth restriction (IUGR) were recorded at delivery. Within each litter, an IUGR piglet is defined as weighing approximately 65% of the birth weight of the largest littermate [26]. The litter weight and average body weight were recorded after farrowing.

Blood samples (5 mL) were collected using vacutainers from the jugular vein of 8 sows per group during farrowing. In addition, 8 neonatal male piglets on the day of birth without ingestion of colostrum were chosen from the above selected 8 sows (1 piglet/sow) [27]. Then, they were anesthetized by the intravenous administration of sodium pentobarbital (50 mg/kg BW) and bled by exsanguination [28]. Following bleeding, 5 mL of blood samples were collected in vacutainers with or without heparin sodium. The blood samples collected without heparin from the heart were centrifuged for 15 min at 3000× *g* to obtain serum, and then all the blood samples were stored at −80 °C until analysis. The jejunum and ileum samples were collected and also stored at −80 °C. Immediately after farrowing, placental tissue samples were collected from similar areas of 16 neonatal boars from sows of the two groups, each group had eight replicates [22].

### 2.4. Sample Analysis

#### 2.4.1. Antioxidant Indicators in Serum of Farrowing Sow, Umbilical Cord, and Neonatal Piglet

The activities of glutathione (GSH), peroxidase (POD), CAT, total antioxidant capacity (T-AOC), SOD, as well as the level of malondialdehyde (MDA) in the serum of farrowing sows, umbilical cords, and neonatal piglets were determined according to the manufacturer’s instructions (Nanjing Jiancheng Biotechnology Institute, Nanjing, China) [29].

#### 2.4.2. Mineral Concentration in Plasma of Farrowing Sow, Umbilical Cord, and Neonatal Piglet

The plasma of farrowing sows, umbilical cords, and neonatal piglets was first diluted with ultrapure water and ultrapure acids (HNO_3_, HClO_4_) to measure the concentration of Ca, P, Mg, Cu, Mn and Zn, then the standard substance was prepared. Plasma samples were prepared with preliminary treatment electric oven digestion under the heating procedure of 80 °C, 60 min; 120 °C, 30 min; and 180 °C, 30 min, and then dried at 260 °C and re-dissolved with 5 mL of 1% HNO_3_. After being transferred into 25 mL volumetric flasks and diluted with 1% HNO_3_ according to the concentration of the samples, all plasma samples were analyzed using an inductively coupled plasma emission spectrometer (IPC 702, Agilent, Santa Clara, CA, USA) following the procedure as described previously [30].

#### 2.4.3. Quantitative Real-Time Transcription PCR (RT-qPCR)

Total RNA was extracted from the placenta, jejunum, and ileum of neonatal piglets with TRIzol reagent and then reverse-transcribed into cDNA using a reverse transcription kit (Takara Biomedical Technology, Kusatsu, Japan). The mRNA expression of genes related to antioxidant function and the transport of mineral elements and *β-Actin* in the placenta, jejunum, and ileum were determined by real-time PCR using SYBR real-time polymerase chain reaction (PCR) (SYBR Premix Ex Taq; Takara Bio Inc., Shiga, Japan). The primer’s information used for RT-qPCR is shown in Table 2. The amplification reactions were carried out in an ABI Prism 7900 HT sequence detection system (Applied Biosystems, Foster, CA, USA). The relative mRNA expression level was calculated using the 2^−ΔΔCt^ method after normalization with *β-Actin* as an internal control [22,30].

### 2.5. Statistical Analysis

The statistical analysis was performed using SPSS 21.0 (2015, IBM-SPSS Inc., Chicago, IL, USA). Data are presented as mean ± standard error of the mean (SEM). A comparison between different groups was performed through independent samples *t*-test. *p* values < 0.05 were considered significant and trends were identified at 0.05 ≤ *p* < 0.10.

## 3. Results

### 3.1. Reproductive Performance of Sows

The effects of maternal CAT supplementation on the reproductive performance of sows are presented in Table 3. Compared with the CON group, dietary CAT supplementation significantly decreased the IUGR rate (*p* < 0.05). For the other reproductive performance indicators, no significant difference between the CON and CAT groups was found.

### 3.2. Antioxidant Enzyme Activities in Serum of Farrowing Sows, Umbilical Cords, and Neonatal Piglets

The effects of maternal CAT supplementation on antioxidant enzyme activities are shown in Figure 1. The results demonstrated that CAT supplementation significantly increased the activity of serum CAT in neonatal piglets and umbilical cords (*p* < 0.05) compared with the CON group, while no significant difference was observed in farrowing sows. The activity of GSH in farrowing sows was also significantly increased (*p* < 0.05) in the CAT supplemented group. Meanwhile, T-AOC in farrowing sows (*p =* 0.089) and umbilical cords (*p =* 0.051) tended to increase in the CAT group compared to the CON group. No significant differences were observed in serum POD, SOD, and MDA concentration between the CON and CAT groups.

### 3.3. mRNA Expression of Genes Related to the Mineral Transport in the Placenta

To gain further insight on the implication of CAT supplementation on placental genes related to the transport of mineral elements, we performed the gene expressions analysis by RT-qPCR (Figure 2). It was revealed that the mRNA expression of *TRPV6* and *CTR1* showed a significant reduction (*p* < 0.05), while the mRNA expression of *ATP2B* tended to decrease (*p =* 0.058) in the CAT group as compared with the CON group.

### 3.4. Effects of CAT on Serum Mineral Elements of Farrowing Sows and Their Neonatal Piglets

The concentrations of mineral elements in the serum of farrowing sows, umbilical cords, and neonatal piglets were measured. As shown in Table 4, the concentration of Ca (*p* = 0.031), Mn (*p* = 0.014) and Zn (*p* = 0.001) in farrowing sows and Mn (*p* < 0.001) in the umbilical cord, as well as Ca (*p* = 0.001), Mg (*p* = 0.008), Cu (*p* = 0.046) and Mn (*p* < 0.001) in neonatal piglets were significantly increased in the CAT supplemented group compared with the CON group. In contrast, the Mg (*p* = 0.042) and Zn (*p* = 0.024) were decreased in the umbilical cord following maternal CAT supplementation.

### 3.5. mRNA Expression of Genes Related to Antioxidant Function in the Placenta, Jejunum, and Ileum

To further assess whether the change in antioxidant enzyme activities in the umbilical cords and piglets was comparable with the change in expression of related genes, we analyzed the expression of *GPX1*, *GPX4*, *CAT*, *Cu/Zn SOD,* and *MnSOD* genes related to antioxidant enzymes. As depicted in Figure 3, it was revealed that maternal CAT supplementation during late pregnancy tended to downregulate (*p =* 0.086) the placental Cu/Zn SOD gene expression, and tended to upregulate the mRNA expression of *GPX1*, *GPX4,* and *CAT* (*p =* 0.084, *p =* 0.063, *p =* 0.052) in the ileum of piglets, while it had no significant effect on the expression of those genes in the jejunum.

## 4. Discussion

### 4.1. Effects of Maternal CAT Supplementation on Reproductive Performance of Sows

Pregnant sows are fed a restricted diet to ensure good health, production, and longevity, but this may lead to starvation stress, which can affect the reproductive performance of sows [31,32,33]. In addition, the rapid development of the fetuses during late gestation may squeeze the digestive tract and cause gastrointestinal dysfunction of sows [34], resulting in elevated systemic oxidative stress. These processes induce sows to produce large amounts of free radicals, resulting in placental dysfunction and fetal growth retardation [35]. Furthermore, the swine intestinal tract is a major target organ of free radical attack, which causes intestinal structure destruction, microbial disturbance, and nutrient absorption obstacles [36]. This may ultimately lead to constipation in prepartum sows and harmful gas generation, which can be absorbed into the bloodstream of the fetus and cause stunted embryo growth. CAT is an antioxidant enzyme against oxidative stress that can inactivate ROS-derived free radicals. In the present study, maternal CAT supplementation significantly decreased the IUGR rate.

IUGR refers to a condition in which the rate of fetal growth is less than normal in light of the growth potential [37]. It reduces neonatal survival, has a permanent stunting effect on postnatal growth and feed efficiency of offspring, and impairs long-term health [38]. Oxidative stress may be a determinant factor leading to complications such as IUGR [39]. A previous study also suggested that IUGR was associated with placental ROS and oxidative injury [40]. Neonates with IUGR were reported to have a low antioxidant defense mechanism, indicating a tight correlation between IUGR and oxidative stress [41]. Developing embryos and fetuses are vulnerable to ROS because they have low levels of most antioxidant enzymes [42]. CAT is one of the ROS-detoxifying enzymes found in the embryo [43]. Abramov et al. (2011) further confirmed that embryopathies were completely blocked by the addition of exogenous CAT and it protected the conceptus from endogenous or exogenous-enhanced oxidative stress [42]. Therefore, it appears that maternal CAT supplementation could decrease the IUGR rate by enhancing the antioxidative capacity of sows and piglets.

However, no significant difference was observed on the other reproductive performance indicators except IUGR in the current study. This is discrepant with the previous study reporting that dietary CAT supplementation reduced birth mortality [13]. This might be related to the difference in the amount and duration of exogenous CAT supplementation.

### 4.2. Maternal CAT Supplementation Regulated Antioxidant Enzyme Activities of Sows and Neonatal Piglets

CAT is one of the crucial antioxidant enzymes that mitigates oxidative stress to a considerable extent by increasing specific enzymatic activity and detoxifying H_2_O_2_ [44,45]. In the present study, the CAT activities in the serum of newborn piglets and umbilical cords were increased after maternal dietary CAT supplementation, suggesting that maternal nutrition may affect oxidative stress in the placenta, and then the fetal redox status is also affected. This is consistent with the previous study reporting that dietary CAT supplementation of sows elevated CAT activity in the plasma of newborn piglets [13]. Likewise, Li et al. [14] reported that dietary supplementation with 2 g/kg exogenous CAT had a beneficial effect on antioxidant capacity in piglets, showing a higher activity of CAT. Furthermore, dietary CAT supplementation markedly increased the GSH activity in the serum of farrowing sows and tended to increase T-AOC levels in the serum of farrowing sows and umbilical cords. These changes are indicative of improved antioxidant function because GSH is an effective antioxidant enzyme in the cells and has well-known protective activity against ROS [46]. Similarly, T-AOC represents the nonenzymatic antioxidant capacity against ROS [47]. Overall, the above results revealed that maternal CAT supplementation enhanced the antioxidant enzyme activities of sows, thereby alleviating oxidative stress in the placenta and fetus.

The maternal CAT supplementation did not affect GSH activity in umbilical cords and neonatal piglets compared to the CON group. Similar results were observed for the POD, SOD, and MDA in the serum of farrowing sows, umbilical cords, and neonatal piglets. These nonsignificant effects of CAT supplementation on the activities of the above antioxidant enzymes may be associated with the fact that CAT can contribute to the reduction in cellular hydroxyl and peroxyl radical load, sequentially decreasing the need for other antioxidant enzymes [48], and potentially demonstrating the more powerful effects of enhanced activity of CAT on the protection of sows against oxidative stress [49]. Furthermore, the effects of dietary CAT supplementation on antioxidant enzyme activities in the plasma of sows may also be related to the duration of sustained ingestion of CAT. Sun et al. [13] observed that sows fed with CAT supplemented diets throughout gestation showed no significant difference in CAT activity during the first 80 days of gestation, while a remarkable rise was observed at day 108. This indicated that dietary CAT supplementation had a considerable effect on some antioxidant enzymes activities of sows as the time elapsed; therefore, it may be reasonable to assume that the CAT activity of farrowing sows did not show a noticeable change after 20 days of exogenous CAT supplementation. These results likely indicate that the mitigation of oxidative stress following dietary CAT supplementation was mainly achieved through catalase rather than other antioxidant enzymes.

### 4.3. mRNA Expression of Genes Related to Mineral Transporters in Placenta

*TRPV6* plays an indispensable role in maternal–fetal calcium transport [50]. It is abundantly expressed in the mammalian placental tissues and functions as an apical Ca entry channel that mediates the transcellular transport of this ion in the placenta [51]. It has been reported that CAT had an inhibitory action on oxidative stress-induced activation of TRP channels [52]. This is consistent with the finding of the present study, which demonstrated that the mRNA expression of *TRPV6* was significantly downregulated following maternal CAT supplementation. Combined with the results of the downward trend of *ATP2B* expression and an increased Ca concentration in the serum of newborn piglets, our results indicated that the utilization of Ca level for newborns may be enough and that additional Ca transporters were not required.

High-affinity Cu transporter (*CTR1*) is a key component of the Cu uptake pathway and peripheral distribution in the placenta [53]. The expression of *CTR1* is affected by Cu concentration. It is expressed in multiple tissues and found in higher levels in perinatal Cu deficiency [54], but excess Cu rapidly induces degradation of *CTR1* [55]. In the present study, we also observed a decreased expression of *CTR1,* which may be due to a significant increase in serum Cu concentration in piglets of the CAT supplemented group.

### 4.4. Effects of CAT on Serum Mineral Elements of Farrowing Sows and Offspring

In the present study, maternal CAT supplementation increased Mg concentration in the serum of newborn piglets, suggesting a relationship between Mg and CAT [56,57]. Mg seems to contribute to the mitigation of oxidative stress due to its antioxidant properties, and its insufficiency can promote ROS production [58,59,60]. Research findings indicated that Mg was involved in more than 600 enzymatic reactions as a cofactor or an activator [61], having an especially wide spectrum of effects in pregnancy [62]. A recent study showed that serum CAT activity was found to be positively correlated with the increase in dietary Mg concentration [63]. A significant positive correlation between Mg concentration and CAT mRNA level in the liver has been reported, suggesting that Mg may participate in the modulation of gene expression of hepatic CAT [64]. Mg deficiency significantly decreased the activity of antioxidant enzymes such as GPX, SOD, and CAT as well as the mRNA expression of *GPX*, *SOD*, and *CAT* genes [65]. In addition, Mg deficiency led to an increase in inflammatory mediators, which are associated with the generation of free radicals [59,65]. Interestingly, the results of the present study also indicated a correlation between antioxidant enzymes and Mg, which might account for the dramatically-increased activity of serum CAT in newborn piglets accompanied by a simultaneous increase in Mg in the serum of neonatal piglets following maternal CAT supplementation.

Trace element Zn is an essential component of the endogenous enzymatic antioxidant defenses such as SOD1 and SOD3. Beyond that, it also has several antioxidant properties including protection of the sulfhydryl groups from oxidative damage [66,67], modulation of intestinal absorption of vitamin E [68], inhibition of ROS production [69], and maintaining the tissue concentration of metallothioneins [70]. In the present study, we found that the concentration of Zn was significantly increased in sows of the CAT supplemented group. These results are in agreement with the previous study on Japanese quail suggesting that supplementation of an antioxidant such as vitamin E improves Zn concentration [71]. The increased concentration of Zn in the serum of sows under oxidative stress may be due to an increase in the absorption of this element following the maternal CAT supplementation. Mn is known to have free radical scavenging capacity without the pro-oxidant side effects of other redox-active metals [72]. The beneficial effect of Mn in antioxidant activity has been reported in the previous study, which showed that Mn ameliorated oxidative stress in rats exposed to chlorpyrifos, an organophosphorus pesticide that can suppress the antioxidant activities of SOD, CAT, and GSH [73]. Exploiting Fe or Mn as a cofactor, CAT catalyzes the degradation or reduction in H_2_O_2_ to produce water and molecular oxygen, indicating that the Mn concentration is positively related to CAT [74,75]. In the present study, the concentration of Mn was increased in farrowing sows, umbilical cords, and neonatal piglets in the CAT supplemented group, and that may contribute to an increase in antioxidant capacity.

It is believed that excessive Ca can lead to inappropriate activation of certain biological processes, resulting in metabolic disorders and cell death. Excessive concentration of intracellular Ca may activate CAT degradation and trigger oxidative stress. It has also been reported that Ca mobilization increases mitochondrial ROS [76]. Antioxidants regulate Ca influx into the cytosol via inhibition of ROS, and CAT prevents mitochondrial permeability transitions by removing mitochondrial H_2_O_2_ [77]. Therefore, it is reasonable to infer that Ca influx can be activated by oxidative stress products and that the elevation of cytoplasmic free Ca can stimulate uptake of mitochondrial Ca. CAT modulates oxidative stress-induced Ca influx. The present study demonstrated that CAT supplementation significantly increased the concentration of Ca in farrowing sows and neonatal piglets, which may be linked to the prevention of Ca influx by CAT and, hence, resulted in an increased concentration of extracellular Ca.

Cu may participate in the formation of ROS and other free radicals [78,79,80], its ability to induce oxidative damage is attributed to the formation of highly reactive hydroxyl radicals (OH^-^) from H_2_O_2_ via the Haber-Weiss reaction [81]. A study in broilers showed that dietary supplementation of Cu at a high concentration (250 mg CuSO_4_/kg diet) resulted in a reduction in the activity of CAT [82]. This phenomenon is consistent with previous results that Cu may inhibit CAT activity by stimulating superoxide radical production [83,84]. However, contradicted results were also observed by others that showed a copper-induced nonsignificant change in CAT activity of zebrafish liver and bass [85,86]. These results suggest that the activity of CAT reduced in response to Cu; however, the extent of reduction varied with Cu concentrations. Additionally, Cu deficiency also increases cellular susceptibility to oxidative damage, adequate levels of Cu are essential for optimal antioxidant defense [78,87]. It has been reported that a deficiency of Cu during pregnancy may lead to oxidative stress [88]. Accordingly, dietary supplementation of Cu nanoparticles enhanced weaned piglets’ antioxidant capacity [89]. In the present study, we observed that CAT supplementation increased serum CAT activity. So, we speculate that CAT may promote the transfer of Cu from sows to their offspring, ensuring adequate levels of Cu are available to the fetus. The increased activity of CAT in neonatal piglets might have some role in the elevated concentration of Cu.

### 4.5. mRNA Expression of Genes Related to Antioxidant Enzymes in the Placenta, Jejunum, and Ileum

Antioxidant enzymes such as SOD, CAT, and GPX have been reported as the first line of defense against free radicles. The two SOD isoforms (Cu/Zn SOD and MnSOD) provide cellular defense against ROS by catalyzing the dismutation of superoxide radicals (O^2−^) to H_2_O_2_, which is further removed by CAT and GPX [90]. The expression of SOD-related genes is a decisive factor for the activity of SOD, but it is also mediated by other factors, such as Zn and Cu levels [91]. As the necessary composition of Cu/Zn SOD, Zn and Cu are closely interrelated to antioxidant functions. In the present study, we observed the mRNA expression of antioxidant-related genes in the placenta and found that the expression of *Cu/Zn SOD* was downregulated in the CAT supplemented group. This might be associated with a significant reduction in Zn concentration in the umbilical cord. Furthermore, we hypothesized that the antioxidant functions of newborn tissues can also be improved; therefore, in the present study, we further analyzed the mRNA expression of related genes in the intestinal tissues. With the upward regulation of expression of *GPX1*, *GPX4* and *CAT* genes, it can be inferred that the maternal CAT supplementation may also be implicated in mitigating the oxidative stress of neonatal piglets.

## 5. Conclusions

Supplementation of CAT in the diet of pregnant sows may decrease the IUGR rate and contribute to improved maternal and fetal antioxidant status, potentially by modulating antioxidant activities, as well as mineral elements in the pregnant sows and their piglets. This finding provides a reasonable foundation for focused intervention studies in the future and may also provide insight into important aspects of mother-infant nutrition in both animals and humans.

## Figures and Tables

**Figure 1 animals-12-00828-f001:**
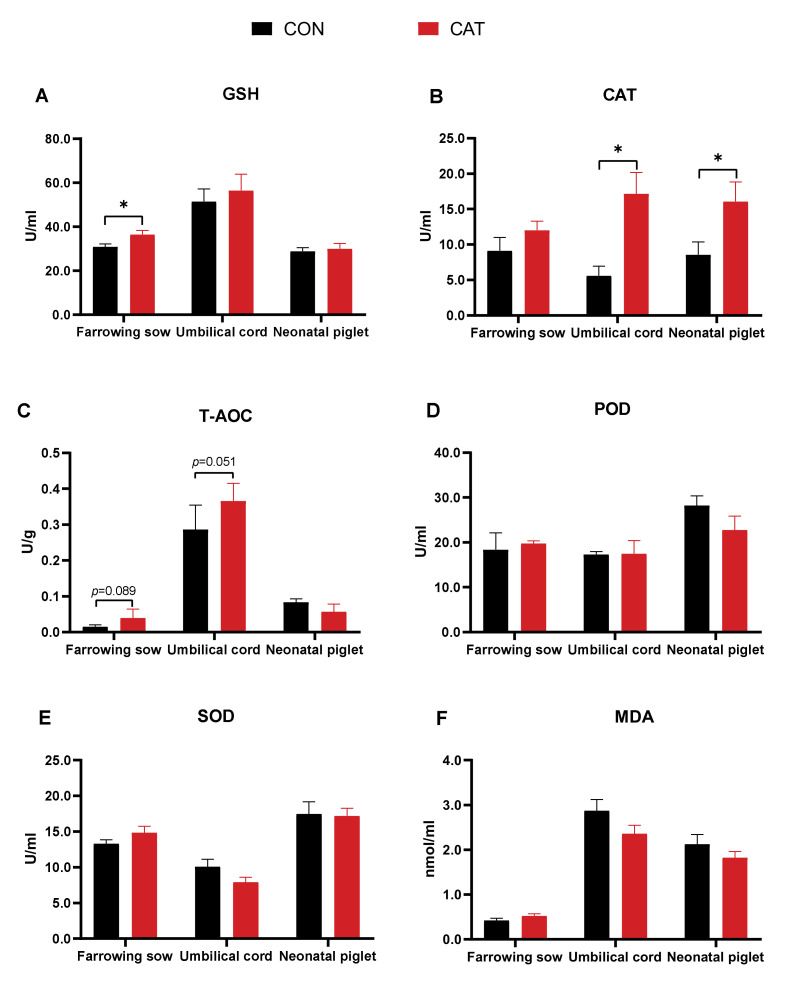
Antioxidant enzyme activities in serum of farrowing sows, umbilical cords, and neonatal piglets. (**A**): glutathione (GSH) activity; (**B**): catalase (CAT) activity; (**C**): Total antioxidant capacity (T-AOC); (**D**): Peroxidase (POD) activity; (**E**) Superoxide dismutase (SOD) activity; (**F**): Malondialdehyde levels (MDA). Data are presented as mean ± SEM, *n* = 8. Statistical significance was set at *p* < 0.05. * Labeled with significant difference (*p* < 0.05).

**Figure 2 animals-12-00828-f002:**
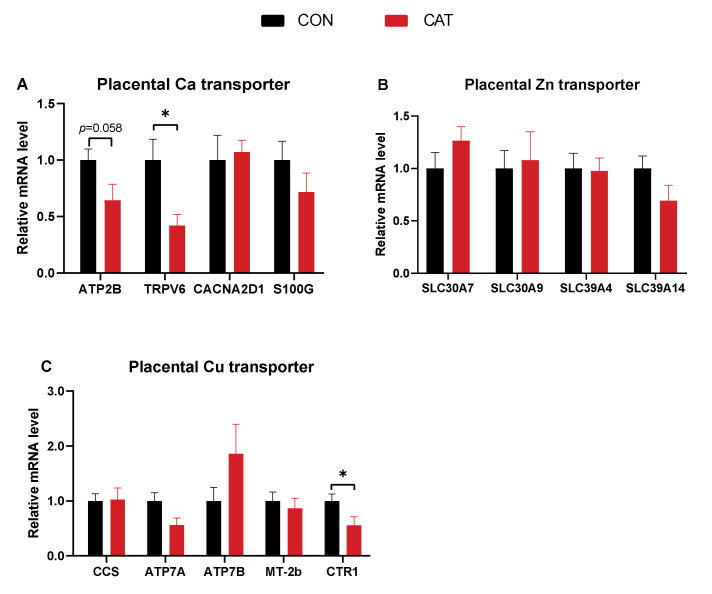
The effects of CAT on the mRNA expression of genes related to mineral transport in the placenta. (**A**): placental Ca transporter; (**B**): placental Zn transporter; (**C**): placental Cu transporter. Data are presented as mean ± SEM, *n* = 8. Statistical significance was set at *p* < 0.05 by *t*-test. * Denotes a significant difference at *p* < 0.05.

**Figure 3 animals-12-00828-f003:**
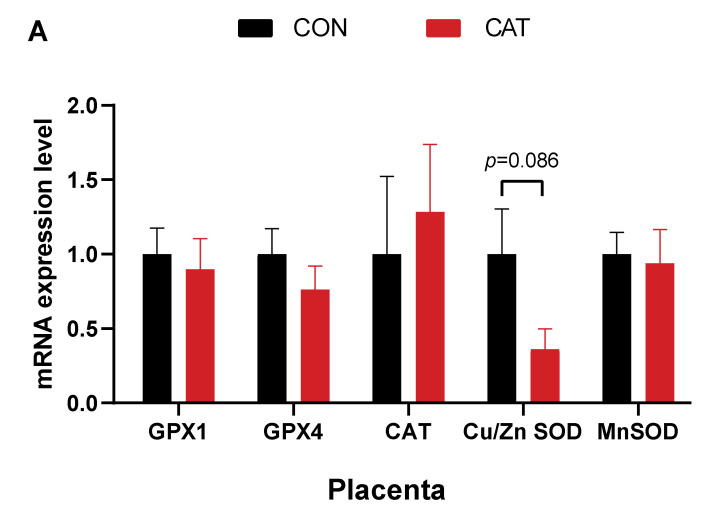
The effects of maternal CAT supplementation on the mRNA expression of genes related to antioxidant function in the placenta, jejunum, and ileum. (**A**): mRNA expression level in the placenta; (**B**): mRNA expression level in the jejunum; (**C**): mRNA expression level in the ileum. Data are presented as mean ± SEM, *n* = 8. Statistical significance was set at *p* < 0.05 by *t*-test.

**Table 1 animals-12-00828-t001:** Composition and nutrient levels of the basal diet (air-dry basis).

Items	Content, %
Ingredient	
Corn	60.6
Soybean meal	11.80
Soybean hull	15.00
Rice bran	7.00
CaHPO4	1.60
Limestone	1.10
Soybean oil	0.80
Acidifier	0.50
Sodium bicarbonate	0.25
Lysine (70%)	0.22
Threonine	0.10
NaCl	0.40
Methionine	0.05
Mold inhibitor	0.08
Choline chloride	0.10
Mineral premix ^1^	0.20
Vitamins premix ^2^	0.20
Total	100
Chemical composition % ^3^	
Digestible energy (Kcal/kg)	3050
Crude protein (%)	12.5
Calcium (%)	0.9
Phosphorus (%)	0.65
Available phosphorus (%)	0.41
Lysine (%)	0.68
Methionine (%)	0.26
Threonine (%)	0.57
Crude fiber (%)	8.0
Crude fat (%)	3.4

^1^ The mineral premix provided the following per kilogram of diet: Copper: 185 mg, Iron: 3500 mg, Manganese: 1250 mg, Zinc: 1300 mg, Selenium: 3.5 mg, Iodine: 190 mg. ^2^ The vitamins premix provided the following per kilogram of diet: Vitamin A: 6000 IU, Vitamin E: 800 mg, Vitamin D3: 4000 IU, Vitamin K3: 30 mg, Vitamin: B1 25 mg, Vitamin B2: 25 mg, Vitamin B6: 80 mg, Niacin 300 mg, Vitamin B12: 0.2 mg, Pantothenic acid: 200 mg, Folic acid: 10 mg, Biotin: 4 mg, Choline: 5000 mg. ^3^ The calculated value of dietary nutrients.

**Table 2 animals-12-00828-t002:** Primers used for RT-qPCR.

Target Gene	Accession NO.	Nucleotide Sequence of Primer (5′-3′)	Size (bp)
*ATP2B*	XM_021091182.1	F: CTGGTTGGATTGAAGGTGCT	123
		R: GCTCCTGCTCAATTCGACTC	
*TRPV6*	XM_021078898.1	F: CTAACAAGCTGGGCCATTTC	119
		R: GCTGTACATGAAGGGCAGGT	
*CACNA2D1*	XM_021102233.1	F: TGTACCTGGATGCACTGGAA	122
		R: TCCCATCACACCAAGAATCA	
*S100G*	NM_214140.2	F: TCCTGCAGAACTGAAGAGCA	133
		R: TAGGGTTCTCGGACCTTTCA	
*SLC30A7*	XM_005655460.2	F: CCTCTTTAACGGTGCTCTCG	119
		R: CATGAAAGTGTCCGTGTCCA	
*SLC30A9*	NM_001137632.1	F: ATTAGGCGTGGTCTCAGCAT	119
		R: TTACTGACGGGTCGTTCTCC	
*SLC39A4*	XM_021090449.1	F: AGCTCAGCCAGTCAGAGAGG	123
		R: TGACGTAGTGGGTAGCAGCA	
*SLC39A14*	XM_005657235.3	F: AGGATGAAAGGAAGGGCAGT	114
		R: TACCCGATCTGGATCTGTCC	
*CCS*	NM_001001866.1	F: CTTCAGGATGGAGGATGAGC	119
		R: TCCCGGTGATCTTGGATAAG	
*ATP7A*	XM_013990938.2	F: TCTGGCAGCACTGTTATTGC	116
		R: GCCTCCTCCACAAGTTTGAC	
*ATP7B*	XM_021065286.1	F: TATGACCCTTCCTGCGTCTC	121
		R: ACCTGGCATCTGTTCCTGTC	
*MT-2b*	XM_003355808.4	F: TCCTGCAAATGCAAAGACTG	119
		R: CACTTGTCCGAGGCTCCTT	
*CTR1*	NM_214100.3	F: CGCAAATCACAAGTCAGCAT	129
		R: CACTGTCTGCAGGAGGTGAG	
*GPX1*	NM_214201.1	F: TGGGGAGATCCTGAATTG	184
		R: GATAAACTTGGGGTCGGT	
*GPX4*	NM_214407.1	F: GATTCTGGCCTTCCCTTGC	173
		R: TCCCCTTGGGCTGGACTTT	
*CAT*	XM_021081498.1	F: CGAAGGCGAAGGTGTTTG	370
		R: AGTGTGCGATCCATATCC	
*Cu/Zn SOD*	NM_001190422.1	F: CCAGTGCAGGTCCTCACTTCAATC	172
		R: CGGCCAATGATGGAATGGTCTCC	
*MnSOD*	NM_214127.2	F: GGACAAATCTGAGCCCTAACG	159
		R: CCTTGTTGAAACCGAGCC	
*β-actin*	XM_003357928.4	F: CGTTGGCTGGTTGAGAATC	132
		R: CGGCAAGACAGAAATGACAA	

*ATP2B* = plasma membrane calcium ATPase; *TRPV6* = Transient Receptor Potential Cation Channel Subfamily V Member 6; *CACNA2D1* = Calcium Voltage-Gated Channel Auxiliary Subunit Alpha2delta 1; *S100G* = S100 Calcium Binding Protein G; *SLC30A7* = Solute Carrier Family 30 Member 7; *SLC30A9* = Solute Carrier Family 30 Member 9; *SLC39A4* = Solute Carrier Family 30 Member 4; *SLC39A14* = Solute Carrier Family 30 Member 14; *CCS* = Copper Chaperone For Superoxide Dismutase; *ATP7A* = ATPase Copper Transporting Alpha; *ATP7B* = ATPase Copper Transporting Beta; *MT-2b* = Metallothionein-like protein 2B; *CTR1* = High affinity copper uptake protein 1; *GPX1* = glutathione peroxidase 1; *GPX4* = glutathione peroxidase 4; *CAT* = catalase; *Cu/Zn SOD* = Copper-zinc-superoxide dismutase; *MnSOD* = Manganese superoxide dismutase.

**Table 3 animals-12-00828-t003:** Effects of maternal CAT supplementation during late pregnancy on the reproductive performance of sows.

Items	Dietary Pretreatment		*p*-Value
	CON Group	CAT Group	
Total born	12.31 ± 0.98	12.31 ± 0.67	1.00
Born alive	11.00 ± 0.77	11.08 ± 0.58	0.94
Birth (alive) litter weight (kg)	15.26 ± 1.02	15.82 ± 0.66	0.65
Mean weight of born alive per piglet (kg)	1.40 ± 0.07	1.47 ± 0.09	0.57
Birth mortality (%) ^1^	8.50 ± 0.03	8.59 ± 0.03	0.98
IUGR rate (%) ^2^	3.89 ± 0.02 ^b^	0.85 ± 0.03 ^a^	0.03

Data are presented as mean ± SEM, *n* = 8. Statistical significances were set at *p* < 0.05 by *t*-test. Means denoted by different superscript letters showed a significant difference at *p* < 0.05. ^1^ Birth mortality (%) = (total born–born alive)/total × 100; ^2^ IUGR rate (%) = the number of IUGR/total × 100.

**Table 4 animals-12-00828-t004:** The effects of maternal CAT supplementation on the serum mineral elements of farrowing sows, umbilical cords, and neonatal piglets.

Items	Dietary Treatment		*p*-Value
	CON Group	CAT Group	
Sow Serum on Farrowing Day			
Ca (ug/mL)	256.05 ± 19.04 ^b^	392.24 ± 45.62 ^a^	0.031
P (ug/mL)	411.99 ± 28.10	411.88 ± 17.99	0.997
Mg (mmol/L)	0.66 ± 0.09	0.51 ± 0.12	0.363
Cu (ug/mL)	6.04 ± 0.28	6.38 ± 0.38	0.480
Mn (ug/mL)	0.12 ± 0.06 ^b^	0.30 ± 0.03 ^a^	0.014
Zn (ug/mL)	0.60 ± 0.09 ^b^	2.93 ± 0.43 ^a^	0.001
Umbilical Cord Serum			
Ca (ug/mL)	774.44 ± 166.77	783.43 ± 41.38	0.951
P (ug/mL)	365.46 ± 53.57	291.37 ± 23.87	0.214
Mg (mmol/L)	0.86 ± 0.09 ^a^	0.53 ± 0.10 ^b^	0.042
Cu (ug/mL)	0.93 ± 0.21	0.98 ± 0.20	0.874
Mn (ug/mL)	0.22 ± 0.03 ^b^	1.14 ± 0.03 ^a^	0.000
Zn (ug/mL)	7.42 ± 0.82 ^a^	5.24 ± 0.37 ^b^	0.024
Neonatal Piglet Serum			
Ca (ug/mL)	316.06 ± 21.28 ^b^	646.63 ± 71.66 ^a^	0.001
P (ug/mL)	270.03 ± 34.04	339.89 ± 46.46	0.258
Mg (mmol/L)	0.58 ± 0.03 ^b^	0.75 ± 0.05 ^a^	0.008
Cu (ug/mL)	0.47 ± 0.10 ^b^	0.81 ± 0.12 ^a^	0.046
Mn (ug/mL)	0.06 ± 0.01 ^b^	0.97 ± 0.05 ^a^	0.000
Zn (ug/mL)	3.39 ± 0.46	4.35 ± 1.34	0.460

Data are presented as mean ± SEM, *n* = 8. Statistical significances were set at *p* < 0.05 by *t*-test. Means denoted by different superscript letters showed a significant difference at *p* < 0.05.

## Data Availability

Data is contained in the article or available from the corresponding author upon reasonable request.

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
