# Peer review of "Effect of Maternal Catalase Supplementation on Reproductive Performance, Antioxidant Activity and Mineral Transport in Sows and Piglets"

_animals, 2022, doi:10.3390/ani12070828_

Round 1
Reviewer 1 Report
This is an interesting topic because the effect of oxidative stress on sow reproductive performance and piglet health are of utmost importance for the swine industry. However, the paper is poorly written and appears as a first draft. The authors should have taken the time to review and proofread it before submission.
Here are some of my comments to improve the contents of this paper:
Abstract:
Define CAT, T-AOC, GXP in the abstract.
Line 89: as defined later in the text, P = 0.085 is a tendency, and it should be specified that CAT tended to improve T-OAC.
Line 42: remove of before mRNA.
Introduction:
Line 55: what is “stressful problem”; simply replace by stress.
Throughout the introduction, some of the abbreviations are italicized and some are not. This must be consistent through the document.
Line 68-73: when you said “several studies…..hepatic apoptosis” you should provide more than 1 reference.
Line 75: the immune system will be reduced instead of “the immune system will reduced”
Line 80: provide references for this assertion “based on previous studied” because the reader doesn’t know which studies you are referring to.
Material and methods:
Line 64: 20 repetitions? This should be 20 replicates.
Line: 100: you mentioned that CAT was supplemented at 2.0 g/d. How was it provided? Maybe express this as g/kg of diet for more clarity.
Sows was fed CAT from d 95 to 110, did you collect blood before farrowing? It should have been interesting to see the changes in antioxidant capacity over time to farrowing.
Lines 108,110, and 113: what are the superscripts referring to in the table 1?
Line 116: for sows, use farrowing instead of delivery.
Line 123: eight duplicate or eight replicate?
Line 133-136: the mineral should be mentioned somewhere in the text. The first time the reader sees them is in the table. In addition, the method explained here is confusing. More detail description should be provided.
Line 139-147: similar to mineral, the authors should mention and define the genes in the text.
Line 149: what is the superscript referring to in the table 2?
Results:
Figure 2: on the graphs, it should be transporter not “transpoter”.
Line 189-194: this is the first time that ileum is mentioned in the text. This should be explained in the material and method (sample collection and sample analysis).
Anywhere you have multiple graphs on a figure, you should name the graph and explain them in the footnote. For example:
Figure 1: explain in the footnote A, B, C, and D that you put on the graphs.
Figures 2 and 3: name the graph and give explanation in the footnote
Discussion:
In general, I don’t see much in the discussion except some literature review at the beginning of each paragraph. The authors should contrast their findings with other reports in the literature and explain or discuss their significant.
Line 224-230: this sentence is long and confusing. I don’t get the message. Rephrase for more clarity.
Line 233-235: the point of this sentence is that the supplementation of CAT did not affect GSH in umbilical cord, or POD, SOD, and MDA in sows, umbilical cord, and neonate piglets. Please, rephrase your sentence for more clarity.
Line 257: remove was before tented.
This paragraph (line 267-281) goes into a review of the literature and discussed very little the result of the current study. Please rewrite this paragraph and base the discussion on the findings of the current study.
This paragraph (line 312-321) doesn’t make sense. Although the authors showed here the role of copper in biological functions, it is not clear how copper is related to oxidative stress and Cat activity. The authors should discuss the effects of CAT supplementation on copper levels and find studies in the literature to contrast their findings. The reference 55 is on plants, but I believe there are studies on the supplementation of CAT in swine that the authors can refer to in the literature.
Line 336-338: why mention growth and development of piglets I the study didn’t go further and no growth parameters were measured?
Conclusion:
Line 242-345: how will the results of this study help improve production performance of sows? You didn’t measure anything related to production the current study.
References did not follow the journal recommendations.
Author Response
Comments and Suggestions for Authors
This is an interesting topic because the effect of oxidative stress on sow reproductive performance and piglet health are of utmost importance for the swine industry. However, the paper is poorly written and appears as a first draft. The authors should have taken the time to review and proofread it before submission.
Response: We thank the reviewer for his/her valuable suggestions. The revised manuscript has been thoroughly revised and issues related to sentence structure, grammar, and spelling errors have been fixed, with a hope that now it can meet the journal’s standard.
Here are some of my comments to improve the contents of this paper:
Abstract:
Define CAT, T-AOC, GXP in the abstract.
Response: Thanks for your useful suggestion. We have defined CAT, T-AOC, GPX1 and GPX4 in the abstract of the revised manuscript.
Line 89: as defined later in the text, P = 0.085 is a tendency, and it should be specified that CAT tended to improve T-OAC.
Response: Thanks for your nice suggestion. We made the suggested changes and marked it with yellow color in the revised manuscript.
Line 42: remove of before mRNA.
Response: Corrected as suggested. Thanks.
Introduction:
Line 55: what is “stressful problem”; simply replace by stress.
Response: Corrected as suggested. Thanks.
Throughout the introduction, some of the abbreviations are italicized and some are not. This must be consistent through the document.
Response: We appreciate the concern of the reviewer. However, it is highlighted that only names of genes are italicized and those of enzymes and proteins are in normal font. We have carefully checked and made changes where required. Hope this concern is addressed.
Line 68-73: when you said “several studies…..hepatic apoptosis” you should provide more than 1 reference.
Response: Thanks for highlighting this oversight on our part. We have now added two more references in the revised MS.
Line 75: the immune system will be reduced instead of “the immune system will reduced”
Response: Corrected as suggested. Thanks.
Line 80: provide references for this assertion “based on previous studied” because the reader doesn’t know which studies you are referring to.
Response: Corrected as suggested. Thanks.
Material and methods:
Line 64: 20 repetitions? This should be 20 replicates.
Response: Corrected as suggested. Thanks.
Line: 100: you mentioned that CAT was supplemented at 2.0 g/d. How was it provided? Maybe express this as g/kg of diet for more clarity.
Response: Thanks for your useful suggestion. We have replaced “2.0 g/d” into “330mg/kg” and marked it with yellow color in the revised manuscript.
Sows was fed CAT from d 95 to 110, did you collect blood before farrowing? It should have been interesting to see the changes in antioxidant capacity over time to farrowing.
Response: Thanks for your useful suggestion. Your suggestion points out a meaningful direction for our follow-up research. We have collected blood before farrowing, which could be used to see the changes in antioxidant capacity over time to farrowing.
Lines 108,110, and 113: what are the superscripts referring to in the table 1?
Response: Apologies for this mistake. We have clarified in the revised version.
Line 116: for sows, use farrowing instead of delivery.
Response: Corrected as suggested. Thanks.
Line 123: eight duplicate or eight replicate?
Response: Corrected as suggested. Thanks.
Line 133-136: the mineral should be mentioned somewhere in the text. The first time the reader sees them is in the table. In addition, the method explained here is confusing. More detail description should be provided.
Response: Thanks for your useful suggestion. Mineral refers to Ca, P, Mg, Cu, Mn and Zn. The roles of these elements have been described and discussed in the discussion chapter. While we have introduced their names in the last paragraph of the introduction chapter and added a more detailed description of mineral element determination, and marked it with yellow color in the revised manuscript.
Line 139-147: similar to mineral, the authors should mention and define the genes in the text.
Response: Thanks for your useful suggestion. Several classical genes related to antioxidant function and transport of minerals were studied in the present manuscript. We have briefly described the functions and roles of these genes in the discussion chapter under respective sub-headings. We also added more details about the genes mentioned in the revised version.
Line 149: what is the superscript referring to in the table 2?
Response: We have removed unnecessary superscripts in Table 2. Thanks.
Results:
Figure 2: on the graphs, it should be transporter not “transpoter”.
Response: Thanks for your careful read and hard work on the previous manuscript. We have changed the ‘transpoter’ to ‘transporter’ in figure 2.
Line 189-194: this is the first time that ileum is mentioned in the text. This should be explained in the material and method (sample collection and sample analysis).
Response: Thanks for your useful suggestion. We have supplemented the material and method for the ileum sample collection and analysis.
Anywhere you have multiple graphs on a figure, you should name the graph and explain them in the footnote. For example:
Figure 1: explain in the footnote A, B, C, and D that you put on the graphs.
Figures 2 and 3: name the graph and give explanation in the footnote
Response: Thanks for your useful suggestion. We have made necessary changes and now the clarity has been optimized. Meanwhile, we have already mentioned self-explanatory titles on the graphs of figure 2 and 3.
Discussion:
In general, I don’t see much in the discussion except some literature review at the beginning of each paragraph. The authors should contrast their findings with other reports in the literature and explain or discuss their significant.
Response: We thank the reviewer for his/her comments and suggestions. We have made some corresponding changes in the revised version.
Line 224-230: this sentence is long and confusing. I don’t get the message. Rephrase for more clarity.
Response: Thanks for your useful suggestion. We have made some corresponding changes and marked it with yellow color in the revised manuscript. Thanks again!
Line 233-235: the point of this sentence is that the supplementation of CAT did not affect GSH in umbilical cord, or POD, SOD, and MDA in sows, umbilical cord, and neonate piglets. Please, rephrase your sentence for more clarity.
Response: Thanks for your useful suggestion. We have rephrased the sentence and marked it with yellow color in the revised manuscript.
Line 257: remove was before tented.
Response: Corrected as suggested. Thanks!
This paragraph (line 267-281) goes into a review of the literature and discussed very little the result of the current study. Please rewrite this paragraph and base the discussion on the findings of the current study.
Response: Thanks for highlighting point. We have made necessary changes in the expression of text to optimize the clarity. We have rewritten most of this paragraph and replaced some particularly old studies with current findings. Based on these new researches, we improved the discussion and updated the references. These modifications were marked with yellow color in the revised version.
This paragraph (line 312-321) doesn’t make sense. Although the authors showed here the role of copper in biological functions, it is not clear how copper is related to oxidative stress and Cat activity. The authors should discuss the effects of CAT supplementation on copper levels and find studies in the literature to contrast their findings. The reference 55 is on plants, but I believe there are studies on the supplementation of CAT in swine that the authors can refer to in the literature.
Response: Thanks for your useful suggestion. We have rewritten this paragraph according to the reviewer’s comment. We provided more details to describe this mechanism of the effects that CAT supplementation and copper levels. We also found relevant studies to contrast our findings. These modifications were marked with yellow color in the revised version. We are grateful for the suggestion.
Line 336-338: why mention growth and development of piglets I the study didn’t go further and no growth parameters were measured?
Response: Thanks for your useful suggestion. It was an inference based on improved antioxidant capacity of newborn piglets. To tone down, we have deleted the sentence about growth and development of piglets.
Conclusion:
Line 242-345: how will the results of this study help improve production performance of sows? You didn’t measure anything related to production the current study.
Response: Thanks for your useful suggestion. It has been corrected in the revised version.
References did not follow the journal recommendations.
Response: Thanks for your useful suggestion. We have checked all the references and corrected them according to the journal style in the revised version.

Reviewer 2 Report
general edits
line 29 edit to studies in "the" future
line 33 remove "and" after expression
line 42 remove or after downregulated
line 44 please indicate if the ilium tissues is from Piglets or sows.
line 62 remove "a" after That
line 75 make utilize past tense- and change "will reduce" to is reduced
line 115 remove samples after blood.
line 169 add space between sows and (
In several places in figures the fonts are not the same size.
line 190 ad "s" to Cord
line 202- in what genes was the change in gene expression observed?
line 206- it is jejunum and ileum of piglets? or only ileum?
lines 216-217 is awkward and needs to be edited for clarity
line 222 is the CAT supplementation in Sows or piglets?
line 224 "higher Activity"
line 226 T-AOC levels in serum of Farrowing
lines 227-231 needs editing for clarity
line 223 Change to alleviating
lines 246-249 awkward- needs editing for clarity
lines 252- 254 edit for clarity.
line 277 change is to are
lines 282-287 is a very long sentence- please edit
line 335 remove s from newborns
Comments
Please make sure to describe what the connection is/the significance of investigating the changes in serum minerals and transport genes is in the introduction. This information is missing.
Reword 93- 95 for clarity. It currently reads as if both groups got both treatments.
In the materials and methods, please explain how the CAT was supplemented in the feed.
line 121 Was all blood (piglets and sows) centrifuged to obtain serum
line 122-123 is unclear- what is fleelty? the sentence structure is awkward.
Line 139, was total RNA collected from intestinal tissues of Sows or piglets?
section 3.2- It is unclear of the significance of investigating the mRNA expression of the element transporters. Adding information on this to the introduction would be helpful.
Author Response
Comments and Suggestions for Authors
general edits
line 29 edit to studies in "the" future
Response: Thanks for your useful suggestion. We have corrected it in the revised version.
line 33 remove "and" after expression
Response: Corrected as suggested. Thanks!
line 42 remove or after downregulated
Response: Corrected as suggested. Thanks!
line 44 please indicate if the ilium tissues is from Piglets or sows.
Response: Thanks for your useful suggestion. The ileum tissues were from piglets and we have indicated it in the revised version.
line 62 remove "a" after That
Response: Thanks. The sentence has been optimized.
line 75 make utilize past tense- and change "will reduce" to is reduced
Response: Thanks. It has been modified as per your suggestion in the revised version.
line 115 remove samples after blood.
Response: Corrected as suggested. Thanks!
line 169 add space between sows and (
Response: Corrected as suggested. Thanks!
In several places in figures the fonts are not the same size.
Response: Thanks for your useful suggestion. We had corrected the fonts in the same size.
line 190 ad "s" to Cord
Response: Corrected as suggested. Thanks.
line 202- in what genes was the change in gene expression observed?
Response: Thanks. We have made some corresponding changes and marked them with yellow color in the revised manuscript.
line 206- it is jejunum and ileum of piglets? or only ileum?
Response: We detected the mRNA expression of genes related to antioxidant enzymes in jejunum and ileum of piglets. The results showed that compared with CON group, only the gene expression of ileum in the CAT group showed an upward trend, while the gene expression of jejunum was not statistically significant. We have made some corresponding changes and marked them with yellow color in the revised manuscript. Thanks.
lines 216-217 is awkward and needs to be edited for clarity
Response: Thanks. We have made some corresponding changes and marked them with yellow color in the revised manuscript.
line 222 is the CAT supplementation in Sows or piglets?
Response: Thanks for your useful suggestion. We have verified the references, which stated that the CAT was supplementary fed to sows, similar to our experiment. Sorry for the confusion and we have made some corresponding changes in the revised version.
line 224 "higher Activity"
Response: Corrected as suggested. Thanks.
line 226 T-AOC levels in serum of Farrowing
Response: Corrected as suggested. Thanks.
lines 227-231 needs editing for clarity
Response: Thanks for your useful suggestion. We have made some corresponding changes and marked them with yellow color in the revised version. We are grateful for your insight.
line 223 Change to alleviating
Response: Corrected as suggested. Thanks.
lines 246-249 awkward- needs editing for clarity
Response: We agree with the comment and rewrote the paragraph in the revised manuscript. Thank you for your valuable suggestion.
lines 252- 254 edit for clarity.
Response: Thanks. We have revised this sentence as per your suggestion and marked them with yellow color.
line 277 change is to are
Response: Corrected as suggested. Thanks.
lines 282-287 is a very long sentence- please edit
Response: Thanks for your useful suggestion. We have made some corresponding changes and marked them with yellow color in the revised version.
line 335 remove s from newborns
Response: Corrected as suggested. Thanks.
Comments
Please make sure to describe what the connection is/the significance of investigating the changes in serum minerals and transport genes is in the introduction. This information is missing.
Response: Thanks for your useful suggestion. In the introduction, we have added a description of the connection between serum minerals and CAT, as well as the significance of investigating the changes in minerals and transporters. We have marked the modifications with yellow color in the revised version.
Reword 93- 95 for clarity. It currently reads as if both groups got both treatments.
Response: Thanks for your useful suggestion. We agree with the comment and rewrote the paragraph in the revised manuscript and marked it with yellow color.
In the materials and methods, please explain how the CAT was supplemented in the feed.
Response: Thanks for your useful suggestion. We have added the explanation about the method of CAT supplementation and marked them with yellow color in the revised manuscript.
line 121 Was all blood (piglets and sows) centrifuged to obtain serum
Response: Thanks for your useful suggestion. Actually, only a portion of the blood in this study was centrifuged to obtain the serum, and we had corrected the method in the revised manuscript and marked them with yellow color.
line 122-123 is unclear- what is fleelty? the sentence structure is awkward.
Response: Thanks for your useful suggestion. In order to improve the clarity of the sentence, we have made some corresponding changes and marked them with yellow color in the revised version.
Line 139, was total RNA collected from intestinal tissues of Sows or piglets?
Response: Thanks for your useful suggestion. Jejunum and Ileum tissues were collected from piglets. We have optimized the clarity.
section 3.2- It is unclear of the significance of investigating the mRNA expression of the element transporters. Adding information on this to the introduction would be helpful.
Response: Thanks for your useful suggestion. We have added several information on the introduction according to your advice and which is marked in yellow color in the revised version.

Reviewer 3 Report
Measures enzyme activity, mineral levels and mRNA, but makes no connection to physiological effects, such as birth intervals, birth weight, piglet viability, etc. Without these there is no basis to conclude that there is a "foundation for focused intervention studies in future and can be useful for improving the production performance of sows and may also provide insight into important aspects of mother-infant nutrition in both animals and humans." Abstract is misleading because it refers to 40 sows, but data were only collected from 8 sows per treatment.

Author Response
Comments and Suggestions for Authors
Measures enzyme activity, mineral levels and mRNA, but makes no connection to physiological effects, such as birth intervals, birth weight, piglet viability, etc. Without these there is no basis to conclude that there is a "foundation for focused intervention studies in future and can be useful for improving the production performance of sows and may also provide insight into important aspects of mother-infant nutrition in both animals and humans." Abstract is misleading because it refers to 40 sows, but data were only collected from 8 sows per treatment.
Response: We are highly thankful to the learned reviewer for critical comments and suggestions. Regarding the point of the learned reviewer that there is no connection to physiological effects: we do agree with the opinion of the reviewer and would like to mention that this point is basically a way forward that how our finding will open a new window for further focused studies in this domain. Nevertheless, we have still toned down the expression in the revised version of MS. We have removed word “production performance”.
Similarly, we apologize for not making the number of sows in each treatment group. We have clarified the number in the material and methods section of revised MS. There were twenty sows in each of the two groups.
For the rest of the comments made in the PDF file of our MS: We have complied with all suggestions and corrections in the revised version of MS.
- L36, But only from 8 sows per group, not 20.
Response: Thanks for your useful suggestion. At the beginning of this research, we selected 20 replicates to ensure the smooth progress of the experiment, but only 8 replicates were randomly selected statistically in the formal experiment.
- L39, Tended to improve.
Response: Thanks for your useful suggestion. We had correlated this in the manuscript and marked it with yellow color.
- L42, Tended to downregulate.
Response: Thanks for your useful suggestion. We had correlated this in the manuscript and marked it with yellow color.
- L52, These occur in women at much earlier stages of pregnancy, not the last 20 days.
Response: Thank you very much for your useful suggestion, indeed the whole paper is focused on late pregnancy research, therefore we had changed the ‘spontaneous abortion and recurrent pregnancy loss’ to ‘idiopathic recurrent pregnancy loss and embryogenesis defect’, and marked them with yellow color in the manuscript.
- L54, In earlier stages of pregnancy.
Response: Thanks for your useful suggestion. We had made the following changes: ‘Increasing oxidative stress in sows especially during late pregnancy could also affect the transport of nutrients and oxygen from the mother to the offspring, thereby inhibiting fetal development’.
- L78, A trimester is 3 months so does not apply to the last third of pregnancy in sows.
Response: Thanks for your useful suggestion. We had changed the ‘in the third trimester of pregnancy of sows’ to ‘during late pregnancy of sows’
- L116, Why only 8 out of 20? How were the sows selected? What stage of delivery? How many piglets born live, dead, mummified per litter?
Response: Thanks for your useful suggestion. At the beginning of this research, we selected 20 replicates to ensure the smooth progress of the experiment, but only 8 replicates were randomly selected statistically in the formal experiment. And we did not have statistics about the stage of delivery and born alive, dead, mummified per litter, because we did not pay more attention to the reproductive performance, etc. in this current research, and thank you very much for the reasonable suggestion about these for our further research.
- L117, How were piglets selected? What was the average birth weight of live pigs in the litters? It does not make sense to talk about effects on fetal development in the introduction and then not measure birth weight.
Response: Thanks for your useful suggestion. In this research, from 16 sows at similar parturient times of two groups, we selected piglets with close to average litter weight that was not breastfed
- L124, Were birth intervals recorded? Was there any effect of supplementation on birth intervals?
Response: Thanks for your useful suggestion. We did not record the birth intervals, but this is of reference significance for our future research. Thank you very much for this question.
- L135, What is the standard substance? What is electric oven digestion?
Response: Thanks for your careful read and hard work on the previous manuscript. We have completed the material and method for the ileum sample collection and analysis. ‘The plasma of farrowing sows, umbilical cords and neonatal piglets was first diluted with ultrapure water and ultrapure acids (HNO3, HClO4) to measure the concentration of calcium (Ca), phosphorus (P), magnesium (Mg), copper(Cu), manganese (Mn), zinc (Zn), and then the standard substance was prepared. Plasma samples were prepared with preliminary treatment electric oven digestion under the heating procedure of 80 ℃, 60 min; 120 ℃, 30 min; and 180 ℃, 30 min, and then dried at 260 ℃ and re-dissolved with 5 ml of 1% HNO3. After being transferred into 25 ml volumetric flasks and diluted with 1% HNO3 according to the concentration of the samples, all plasma samples were analyzed using an inductively coupled plasma emission spectrometer’.
- L245, Does CAT accumulate or is it degraded?
Response: Thanks for your useful suggestion. The serum activity of CAT in sows had no significant difference in the current study.
- L344, There are no indicators whatsoever of production performance or mother-infant nutrition in this study, so there is no foundation for intervention studies.
Response: Thanks for your useful suggestion. We paid more attention on the antioxidant ability and mineral elements changes in this current study based on the mother-child axes, therefore, the production performance or mother-infant nutrition did not analyze currently, but we will notice these indexes for further research, thank you very much for the suggestion again.

Round 2
Reviewer 1 Report
The authors have made effort to respond to my comments. Below are some suggestions to improve the content of the paper.
Line 34: “sow with similar parity of 95 days” is confusing. What is the parity? I assume the experiment started at 95 day of gestation.
Line 36: what about ileum and jejunum? Although, 40 sows were used in the experiment, the data were collected only on 8 of them. I suggest you mention that in the abstract, so we know that your N = 8 per treatment.
Line 83: I don’t think you studied mineral deposition in this experiment. I suggest you just mention mineral transport.
Line 117: remove And.
Line 122: Although T-AOC was defined in the abstract, I think you should define it again here.
Line 139: primer’s information used for RT-qPCR
Line 229: I suggest you change reasonably by “likely”
Line 235-238: this sentence is long and the punctuation makes it difficult to read.
Lines 253 and 255: because magnesium deficiency (MgD) is used only twice in the document and there are so many abbreviations in the paper, I suggest you remove MgD and spell it out in the line 255.
Line 265: the reference 55 is in quail not in swine. So, the author should mention that.
The references:
The authors should revise the reference formatting to align with the journal style (see journal abbreviations).
Author Response
Comments and Suggestions for Authors
The authors have made effort to respond to my comments. Below are some suggestions to improve the content of the paper.
Response: Thank you very much for your previous comments that helped us improve this manuscript.
Line 34: “sow with similar parity of 95 days” is confusing. What is parity? I assume the experiment started at 95 day of gestation.
Response: Thanks for pointing out this. Parity refers to the previous number of farrowing because the pregnant sows we used were not in their first parity. We considered to chose sows with similar parity as parity affects nutritional requirement, milk letdown, and fetal size. We have vividly written the sentence in the revised version to avoid confusion.
Line 36: what about ileum and jejunum? Although, 40 sows were used in the experiment, the data were collected only on 8 of them. I suggest you mention that in the abstract, so we know that your N = 8 per treatment.
Response: Thanks for your useful suggestion. We have made corresponding changes and marked it with green color in the revised manuscript. Thanks again!
Line 83: I don’t think you studied mineral deposition in this experiment. I suggest you just mention mineral transport.
Response: We appreciate the concern of the reviewer. We have carefully checked and made corresponding changes including the title of the paper. Hope this concern is addressed.
Line 117: remove And.
Response: Corrected as suggested. Thanks!
Line 122: Although T-AOC was defined in the abstract, I think you should define it again here.
Response: Thanks for highlighting this oversight on our part. We have defined T-AOC again in the revised version.
Line 139: primer’s information used for RT-qPCR
Response: Corrected as suggested. Thanks.
Line 229: I suggest you change reasonably by “likely”
Response: : Corrected as suggested. Thanks.
Line 235-238: this sentence is long and the punctuation makes it difficult to read.
Response: Thanks for your useful suggestion. In order to improve the clarity of the sentence, we have made corresponding changes and marked them with green color in the revised version. We are grateful for your insight.
Lines 253 and 255: because magnesium deficiency (MgD) is used only twice in the document and there are so many abbreviations in the paper, I suggest you remove MgD and spell it out in the line 255.
Response: Thanks. It has been modified as per your suggestion in the revised version.
Line 265: the reference 55 is in quail not in swine. So, the author should mention that.
Response: Thanks for your useful suggestion. It has been modified as per your suggestion in the revised version.
The references:
The authors should revise the reference formatting to align with the journal style (see journal abbreviations).
Response: Thanks for your useful suggestion. We have checked all the references and corrected them according to the journal style in the revised version.
Reviewer 3 Report
Thank you for all the clarifications in the manuscript. While it is interesting to see the various effects of CAT on indicators of oxidative stress, it is impossible to determine whether any of these effects have any practical significance without information on the sows and piglets, including litter size, duration of farrowing, and birth weight.
Author Response
Comments and Suggestions for Authors
Thank you for all the clarifications in the manuscript. While it is interesting to see the various effects of CAT on indicators of oxidative stress, it is impossible to determine whether any of these effects have any practical significance without information on the sows and piglets, including litter size, duration of farrowing, and birth weight.
Response: Thank you for pointing this out. We agree that this is a potential limitation of the study.
The present study focused on the effect of maternal catalase supplementation on antioxidant activity and mineral transport in sows and piglets. Actually, we also investigated the effects of dietary CAT supplementation on the reproductive performance of sows and piglets, including litter size, duration of farrowing, and birth weight. This would be a meaningful direction for our follow-up research, preferably in conjunction with the changes in intestinal health and microflora in the farrowing sows and neonatal piglets. The results with practical significance will be detailedly presented in our subsequent publications. This is why we have toned down our conclusion during the first round of review.
There's no denying that the present study has opened a new window for further focused studies in this domain. For this reason, mitigation of oxidative stress and associated damages are not only beneficial to animal health but can also improve their production performance. We noted also that before submission of our manuscript there has been a publication by Xiaojiao Sun et at in Animal Bioscience showing that dietary CAT supplementation reduced the duration of parturition, stillbirth, and increased growth performance of piglets. Therefore, we are highly thankful to the learned reviewer for critical comments and suggestions. And we seek the reviewer’s tolerance and understanding.
This manuscript is a resubmission of an earlier submission. The following is a list of the peer review reports and author responses from that submission.